# Seasonality of ventricular fibrillation at first myocardial infarction and association with viral exposure

**Charlotte Glinge**[1,2]*, **Thomas Engstrøm**[1,3], **Sofie E. Midgley**[4], **Michael W. T. Tanck**[5], **Jeppe Ekstrand Halkjær Madsen**[1,6], **Frants Pedersen**[1], **Mia Ravn Jacobsen**[1], **Elisabeth M. Lodder**[2], **Nour R. Al-Hussainy**[1], **Niels Kjær Stampe**[1], **Ramona Trebbien**[4], **Lars Køber**[1], **Thomas Gerds**[6], **Christian Torp-Pedersen**[7,8], **Thea Kølsen Fischer**[4,9], **Connie R. Bezzina**[2], **Jacob Tfelt-Hansen**[1,10], **Reza Jabbari**[1,11]

1 The Heart Centre, Department of Cardiology, Copenhagen University Hospital, Rigshospitalet, Copenhagen, Denmark, 2 Amsterdam UMC, University of Amsterdam, Heart Center, Department of Clinical and Experimental Cardiology, Amsterdam Cardiovascular Sciences, Amsterdam, The Netherlands, 3 Department of Cardiology, University of Lund, Lund, Sweden, 4 Department of Virus and Microbiological Special Diagnostics, Division of Infectious Disease Preparedness, Statens Serum Institut, Copenhagen, Denmark, 5 Amsterdam UMC, University of Amsterdam, Department of Clinical Epidemiology, Biostatistics and Bioinformatics, Amsterdam Public Health (APH), Amsterdam, The Netherlands, 6 Section of Biostatistics, Faculty of Medical Sciences, University of Copenhagen, Copenhagen, Denmark, 7 Department of Clinical Investigation and Cardiology, Nordsjaellands Hospital, Hillerød, Denmark, 8 Department of Cardiology, Aalborg University Hospital, Aalborg, Denmark, 9 Department of Infectious Diseases and Department of Global Health, Clinical Institute, University of Southern Denmark, Odense, Denmark, 10 Department of Forensic Medicine, Faculty of Medical Sciences, University of Copenhagen, Copenhagen, Denmark, 11 International External Collaborator Sponsored Staff at Division of Preventive Medicine, Brigham & Women's Hospital, Boston, MA, United States of America

* cglinge@gmail.com

## Abstract

### Aims

To investigate seasonality and association of increased enterovirus and influenza activity in the community with ventricular fibrillation (VF) risk during first ST-elevation myocardial infarction (STEMI).

### Methods

This study comprised all consecutive patients with first STEMI (n = 4,659; aged 18–80 years) admitted to the invasive catheterization laboratory between 2010–2016, at Copenhagen University Hospital, Rigshospitalet, covering eastern Denmark (2.6 million inhabitants, 45% of the Danish population). Hospital admission, prescription, and vital status data were assessed using Danish nationwide registries. We utilized monthly/weekly surveillance data for enterovirus and influenza from the Danish National Microbiology Database (2010–2016) that receives copies of laboratory tests from all Danish departments of clinical microbiology.

### Results

Of the 4,659 consecutively enrolled STEMI patients, 581 (12%) had VF before primary percutaneous coronary intervention. In a subset (n = 807), we found that VF patients

**Data Availability Statement:** All relevant data are within the manuscript.

**Funding:** This study was supported by funding from the Novo Nordisk Foundation (number

NNFOC140011573), Copenhagen, Denmark, and C. Glinge was funded by the Copenhagen University Hospital, Rigshospitalets Forskningspuljer, Rigshospitalet, Copenhagen, Denmark, and the ESC Research Grant. The funders were not involved in any aspects of the study or the decision to submit for publication.

**Competing interests:** The authors have declared that no competing interests exist.

experienced more generalized fatigue and flu-like symptoms within 7 days before STEMI compared with the patients without VF (OR 3.39, 95% CI 1.76–6.54). During the study period, 2,704 individuals were diagnosed with enterovirus and 19,742 with influenza. No significant association between enterovirus and VF (OR 1.00, 95% CI 0.99–1.02), influenza and VF (OR 1.00, 95% CI 1.00–1.00), or week number and VF (p-value 0.94 for enterovirus and 0.89 for influenza) was found.

## Conclusion

We found no clear seasonality of VF during first STEMI. Even though VF patients had experienced more generalized fatigue and flu-like symptoms within 7 days before STEMI compared with patients without VF, no relationship was found between enterovirus or influenza exposure and occurrence of VF.

## Introduction

Sudden cardiac death (SCD) caused by ventricular fibrillation (VF) during acute myocardial infarction (MI) is a major cause of cardiovascular mortality.[1–3] Susceptibility to VF during acute ischemia is undoubtedly multifactorial and modulated by several factors, including autonomic dysregulation, electrolyte disturbance, hemodynamic dysfunction, inherited factors, and various environmental influences.[1,4,5]

Seasonal variation has been reported in the incidence of sudden cardiac arrest (SCA) and sudden death/SCD,[6–10] typically with a winter peak and a summer nadir. Exposure to cold weather is considered the main factor influencing this seasonality.[7] However, these previous epidemiological studies are limited by differences in the definition of SCD (none had VF recording available) used and the fact that they often included SCD cases occurring in the setting of different cardiac pathologies (i.e., etiological heterogeneity in substrate for VF). Nevertheless, these seasonal patterns indicate the presence of seasonal external factors, such as viral infections, that could trigger VF. While several studies indicate consistent association between viral infections and MI,[11–13] there is weaker evidence of an association between viral exposure and cardiac arrhythmias and SCA[14,15]. A genome-wide association study in the Dutch AGNES (Arrhythmia Genetics in the NEtherlandS) population showed that the most significant association with VF during first ST-elevation myocardial infarction (STEMI) is localized at chromosome 21q21 (SNP: rs2824292, odds ratio = 1.78, 95% CI 1.47–2.13, P = 3.3x10$^{-10}$).[16] Interestingly, the closest gene to this locus is the *CXADR* gene, which encodes the coxsackie (a member of enterovirus (EV)) and adenovirus receptor (CAR-receptor).[16] Furthermore, electrophysiological studies and molecular analyses on CAR in mice showed that CAR is a novel modifier of ventricular conduction and arrhythmia vulnerability in the setting of myocardial ischemia.[17] Moreover, we and others previously showed that SCD victims and individuals resuscitated from SCA experienced generalized fatigue and influenza-like symptoms in the days to weeks before SCD/SCA,[18–20] supporting the role of viral exposure in susceptibility to VF during an acute MI. Several studies have shown that influenza vaccine protects against acute MI and heart failure.[21–23] In addition, because EV and influenza epidemics can be accurately and reliably forecast, such forecasts could advise individuals with chest pain and flu-like symptoms to seek medical care in an early phase during outbreaks.

In this study, the relationship between monthly/weekly incidence of EV or influenza virus and the risk of VF during a first STEMI was investigated by correlating Danish surveillance data of EV and influenza exposure in the general population over a period of almost 7 years with individual data on VF risk before primary percutaneous coronary intervention (PPCI) among first STEMI patients.

## Methods

### Study population

The study population consisted of all consecutive patients aged between 18–80 years with a first STEMI (n = 4,659) admitted to the invasive catheterization laboratory between January 1, 2010 and October 31, 2016 at the Copenhagen University Hospital, Rigshospitalet. Rigshospitalet covers all of eastern Denmark (2.6 million citizens), which corresponds to 45% of the entire Danish population. The STEMI patients were divided in two groups. The main outcome was if VF occurred within the first 12 hours of symptoms of STEMI before PPCI. We included both out-of-hospital and in-hospital cardiac arrest with STEMI and VF (82% out-of-hospital and 17% during emergency care or in hospital (at arrival)). Patients with out-of-hospital cardiac arrest were included on admission to the PCI center after resuscitation by trained emergency medical service personnel. VF had to occur before the guided catheter insertion for PPCI; all VF cases had ECG-documented VF. In addition, all medical reports (emergency and hospital records) and discharge summaries were reviewed to verify patients as VF cases. Patients who were non-Danish citizens (n = 180) were excluded.

The study was approved by the Danish Data Protection Agency (No. 2012-58-0004). Registers were available in an anonymous setup; individual patients were not identifiable because the personal identification numbers were encrypted. We obtained approval from the Danish Patient Safety Authority to gain information on medical reports and discharge summaries (No. 3-3013-2277/1). In Denmark, ethical approval is not required for retrospective, register-based studies.

### Data sources

A unique and permanent personal identification number is assigned to all residents in Denmark. This number is used in all Danish health and administrative registries and enables individual-level linkage between all nationwide registries unambiguously.

Fig 1 shows the data collection. We used data from the electronic Eastern Danish Heart Registry comprising detailed clinical data on all consecutive STEMI patients undergoing cardiac catherization and coronary revascularization in Eastern Denmark. The electronic Eastern Danish Heart Registry includes detailed clinical data on all patients undergoing coronary angiography, such as baseline demographics, angiographic findings, and procedure characteristics from the PPCI.[24] All these data were routinely registered by the PCI operator and assistants in the catheterization laboratory during the PPCI. This registry was further linked to the following three nationwide administrative registries via the personal identification number. (1) The Danish Civil Registration System contains daily changes in the vital status of all residents, and no historical data are deleted.[25] (2) The Danish National Patient Registry holds information on all admissions to hospitals since 1978 and outpatient visits since 1995, coded according to the International Classification of Diseases (ICD)-8 and ICD-10.[26] Information on date of admission, date of discharge, and diagnoses of comorbidities was recorded for every STEMI patient prior to index STEMI. (3) The Danish Registry of Medicinal Product Statistics, which registers all prescriptions dispensed from Danish pharmacies since 1995. Each drug is coded according to the international Anatomical Therapeutic Chemical (ATC) classification system,

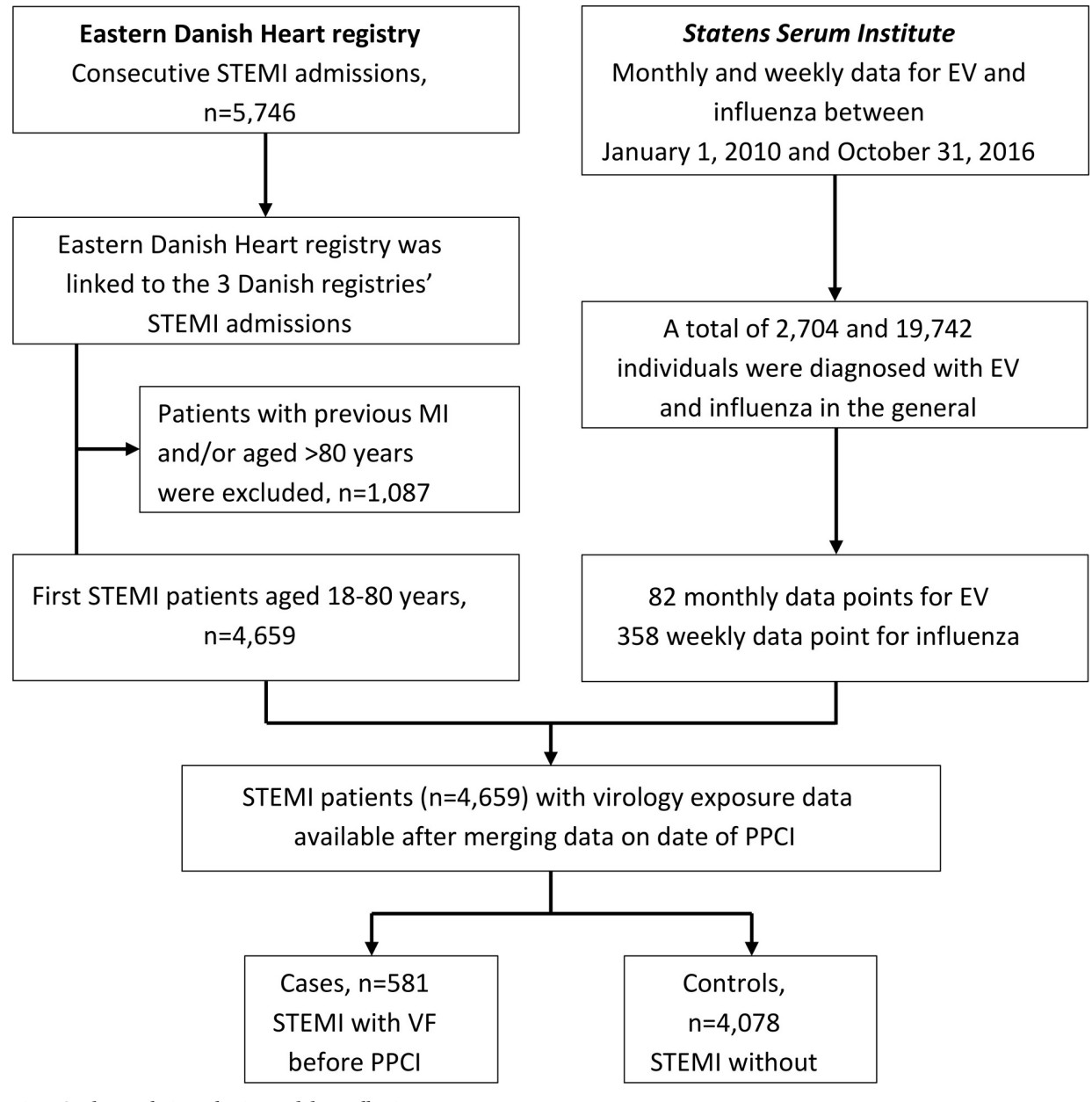

**Fig 1. Study population selection and data collection.**

and the registry includes data on type of drug, strength, quantity dispensed, and date of dispensing [27], which has been shown to be accurate.[28]

Because we were not able to obtain the actual viral exposure on an individual level for the study population, we used monthly and weekly virus incidences at the national level as a proxy for the viral exposure. The virology data were obtained from *Statens Serum Institute*, a Danish governmental public health and research institute operating within the Danish Ministry of Health. We utilized weekly and monthly surveillance data for EV and influenza from the Danish National Microbiology Database (2010–2016), which captures all positive laboratory test results conducted at hospital laboratories in Denmark.[29] This surveillance system monitors national trends in circulating viral infections and collects data on EV and influenza at the

genotype level, as well as patient demographic information and clinical details, including symptoms and date of symptom onset. During the study period of 82 months, 2,704 individuals were diagnosed with EV in the general population. Due to higher incidence of influenza, we calculated the number of influenza events per week, and during the study period of 358 weeks, 19,742 individuals were diagnosed with influenza. The weekly and monthly viral exposure data were then combined and merged with the Eastern Danish Heart Registry using the date of PPCI (Fig 1). The data points comprised monthly and weekly counts of individuals who tested positive for EV and influenza, respectively, obtained from patients who presented to GPs, as well as from hospitalized patients.

## Statistical analysis

Patient characteristics at index STEMI are summarized as the mean and standard deviation (SD) or proportion according to VF status and compared with t-tests and chi-square tests as appropriate.

Separate multivariable logistic regressions were performed for the association of EV and influenza exposure on the day of STEMI and the risk of VF. The models included EV and influenza incidences as continuous variables and were adjusted for week number as a nominal variable with 52 levels and calendar year. Moreover, we performed a full adjusted model, including, week number, calendar year, as well as sex, age, hypertension, atrial fibrillation (AF), chronic kidney disease, Killip class, culprit lesion, preprocedural TIMI-flow, smoking, and previous history of angina. Reported were odds ratios (OR) and accompanying 95% confidence intervals (CI). The likelihood ratio test was used to test the significance of week number.

In a subset of patients (n = 807),[30] we had detailed data on whether the patient had experienced generalized fatigue and flu-like symptoms within the last week prior to the STEMI. A logistic regression model was fitted on this subset. The model was adjusted for week number, calendar year, sex, age, hypertension, AF, chronic kidney disease, Killip class, culprit lesion, preprocedural TIMI-flow, smoking, and previous history of angina.

Analyses were performed using SAS (version 9.4, SAS Institute Inc, Cary, NC, USA) and R. [31]

## Results

### Clinical characteristics

Between January 2010 and October 2016, there were 4,659 consecutively enrolled first STEMI patients (age 18–80 years) at Rigshospitalet in Copenhagen, of whom 581 (12%) had VF before PPCI. The clinical characteristics at the time of STEMI are shown in Table 1. Patients with VF were more likely to be men, be a current smoker, have a history of AF and experience no angina compared with STEMI patients without VF. Furthermore, the proportion on anticoagulant therapy was higher among patients with VF compared with patients without, likely due to the higher degree of AF (Table 1). The angiographic and other presenting characteristics at the time of STEMI according to VF status are outlined in Table 1. Patients with VF had shorter times from symptom onset to PPCI, lower left ventricular ejection fraction, were more likely to have anterior infarctions with left anterior descending artery occlusions, and had lower pre-PCI TIMI flow and higher Killip class.

### Seasonality of infections and VF during first STEMI

During the study period in the general population, 2,704 individuals were diagnosed with EV and 19,742 with influenza. Graphs of seasonal patterns of influenza and EV and number of VF

**Table 1. Patient characteristics at index ST-elevation myocardial infarction according to ventricular fibrillation.**

| Variable | STEMI with VF (n = 581) | STEMI without VF (n = 4078) | P-value |
|---|---|---|---|
| Sex, men, n (%) | 468 (80.6) | 3113 (76.3) | 0.028 |
| Age at STEMI, years, mean (SD) | 59.9 (10.3) | 60.7 (10.9) | 0.079 |
| Age category, n (%) | | | |
| <35 | <5 | 36 (0.9) | 0.374 |
| 35–49 | 110 (18.9) | 697 (17.1) | |
| 50–64 | 263 (45.3) | 1790 (43.9) | |
| > = 65 | 205 (35.3) | 1555 (38.1) | |
| BMI, kg/m$^2$, mean (SD) | 27.9 (14.5) | 27.3 (9.9) | 0.267 |
| Missing, n | 87 | 120 | |
| Smoking, n (%) | | | |
| Current | 285 (56.8) | 1899 (51.1) | 0.025 |
| Never | 96 (19.1) | 894 (24.1) | |
| Past | 121 (24.1) | 921 (24.8) | |
| Missing, n | 79 | 364 | |
| Family history of ischemic heart disease, n (%) | 190 (70.1) | 2312 (65.5) | 0.143 |
| Missing, n | 310 | 550 | |
| Canadian Cardiovascular Society (CCS) grading of angina pectoris, n (%) | | | |
| CCS I-II | 33 (7.6) | 445 (11.2) | 0.032 |
| CCS III-IV | 12 (2.8) | 72 (1.8) | |
| No angina | 390 (89.7) | 3462 (87.0) | |
| Missing, n | 146 | 99 | |
| **COMORBIDITIES** | | | |
| Hypertension, n (%) | 100 (17.2) | 620 (15.2) | 0.233 |
| Diabetes, n (%) | 40 (6.9) | 328 (8.0) | 0.375 |
| Atrial fibrillation, n (%) | 33 (5.7) | 102 (2.5) | < 1e-04 |
| Hypercholesterolemia, n (%) | 26 (4.5) | 238 (5.8) | 0.218 |
| Peripheral vascular disease, n (%) | 22 (3.8) | 103 (2.5) | 0.105 |
| Stroke, n (%) | 19 (3.3) | 122 (3.0) | 0.812 |
| Congestive heart failure, n (%) | 15 (2.6) | 86 (2.1) | 0.562 |
| Chronic kidney disease, n (%) | 10 (1.7) | 98 (2.4) | 0.382 |
| Liver disease, n (%) | 9 (1.5) | 61 (1.5) | 1.000 |
| Malignancy, n (%) | 39 (6.7) | 367 (9.0) | 0.080 |
| **PHARMACOTHERAPY—3 months before** | | | |
| Renin-angiotensin-system blockers, n (%) | 113 (19.4) | 728 (17.9) | 0.379 |
| Statins, n (%) | 63 (10.8) | 444 (10.9) | 1.000 |
| Calcium channel blockers, n (%) | 61 (10.5) | 440 (10.8) | 0.889 |
| Antidiabetics, n (%) | 40 (6.9) | 351 (8.6) | 0.187 |
| Beta-blockers, n (%) | 59 (10.2) | 328 (8.0) | 0.100 |
| Acetylsalicylic acid, n (%) | 42 (7.2) | 290 (7.1) | 0.987 |
| Diuretics (combi), n (%) | 37 (6.4) | 254 (6.2) | 0.969 |
| Thiazide, n (%) | 35 (6.0) | 233 (5.7) | 0.837 |
| Loop diuretics, n (%) | 20 (3.4) | 98 (2.4) | 0.177 |
| Potassium supplements, n (%) | 16 (2.8) | 85 (2.1) | 0.376 |
| Anti-coagulantia, n (%) | 26 (4.5) | 69 (1.7) | < 1e-04 |
| Anti-adrenergic drugs, n (%) | <5 | 58 (1.4) | 0.211 |
| Spironolactone, n (%) | 6 (1.0) | 26 (0.6) | 0.418 |
| **PPCI VARIABLES** | | | |

*(Continued)*

**Table 1.** (Continued)

| Variable | STEMI with VF (n = 581) | STEMI without VF (n = 4078) | P-value |
|---|---|---|---|
| Time from symptom onset to PPCI, min, mean (SD) | 172.0 (146.8) | 268.1 (503.8) | < 1e-04 |
| Missing, n | 30 | 93 | |
| Pre-procedural LVEF, n (%) | | | |
| LVEF < = 35 | 142 (24.4) | 238 (5.8) | < 1e-04 |
| LVEF >35 | 189 (32.5) | 895 (21.9) | |
| Missing | 250 (43.0) | 2945 (72.2) | |
| Infarct location, n (%) | | | |
| Anterior | 262 (45.1) | 1616 (39.6) | < 1e-04 |
| Non-anterior | 212 (36.5) | 2213 (54.3) | |
| Missing, n | 107 | 249 | |
| Culprit lesion, n (%) | | | |
| LAD | 302 (54.1) | 1732 (42.6) | < 1e-04 |
| Non-LAD | 256 (45.9) | 2336 (57.4) | |
| Missing | 23 | 10 | |
| Pre-procedural TIMI flow, n (%) | | | |
| TIMI 0-I | 21 (3.8) | 87 (2.2) | 0.015 |
| TIMI II-III | 172 (31.2) | 1478 (36.6) | |
| Missing, n | 30 | 40 | |
| Killip class, n (%) | | | |
| Killip Class I | 454 (83.0) | 3711 (94.6) | < 1e-04 |
| Killip Class >I | 93 (17.0) | 213 (5.4) | |
| Missing, n | 34 | 154 | |

cases in the setting of first STEMI over the study time are shown in Fig 2. Influenza showed strong winter and spring seasonality (week 40 to week 20), with a maximum observed during the first 10 weeks of the year. EV was detected year-round but tended to peak, as expected, during late summer and fall months. There was no clear seasonality of VF during STEMI.

Results from the logistic regression analyses for VF before PPCI are summarized in Fig 3. We found no association between EV and VF (OR 1.00, 95% CI 0.99–1.02) or influenza and VF (OR 1.00, 95% CI 1.00–1.00). We also found no association between week number and VF (p-value 0.94 for EV model and 0.89 for influenza model), confirming the lack of seasonality in the risk of VF found in Fig 2.

### Subanalysis–generalized fatigue and flu-like symptoms prior to event

In a subset of the patients (n = 807),[30] we had detailed data on whether the patient had experienced generalized fatigue and flu-like symptoms within the last week prior to the STEMI. Compared with the STEMI patients without VF prior to PPCI (n = 520), the VF cases (n = 287) had significantly more generalized fatigue and flu-like symptoms within 7 days before STEMI (14% vs. 5%, p<0.0001, OR 3.39, 95% CI 1.76–6.54). In addition, the majority of the symptoms were reported for STEMI patients during influenza season (defined as week 40 to week 20) (76% vs. 24%, p = 0.024).

## Discussion

### Main findings

This cohort study is the first to investigate the relationship between viral exposure and VF risk at first STEMI before PPCI. The study consecutively enrolled 4,659 patients with STEMI (aged

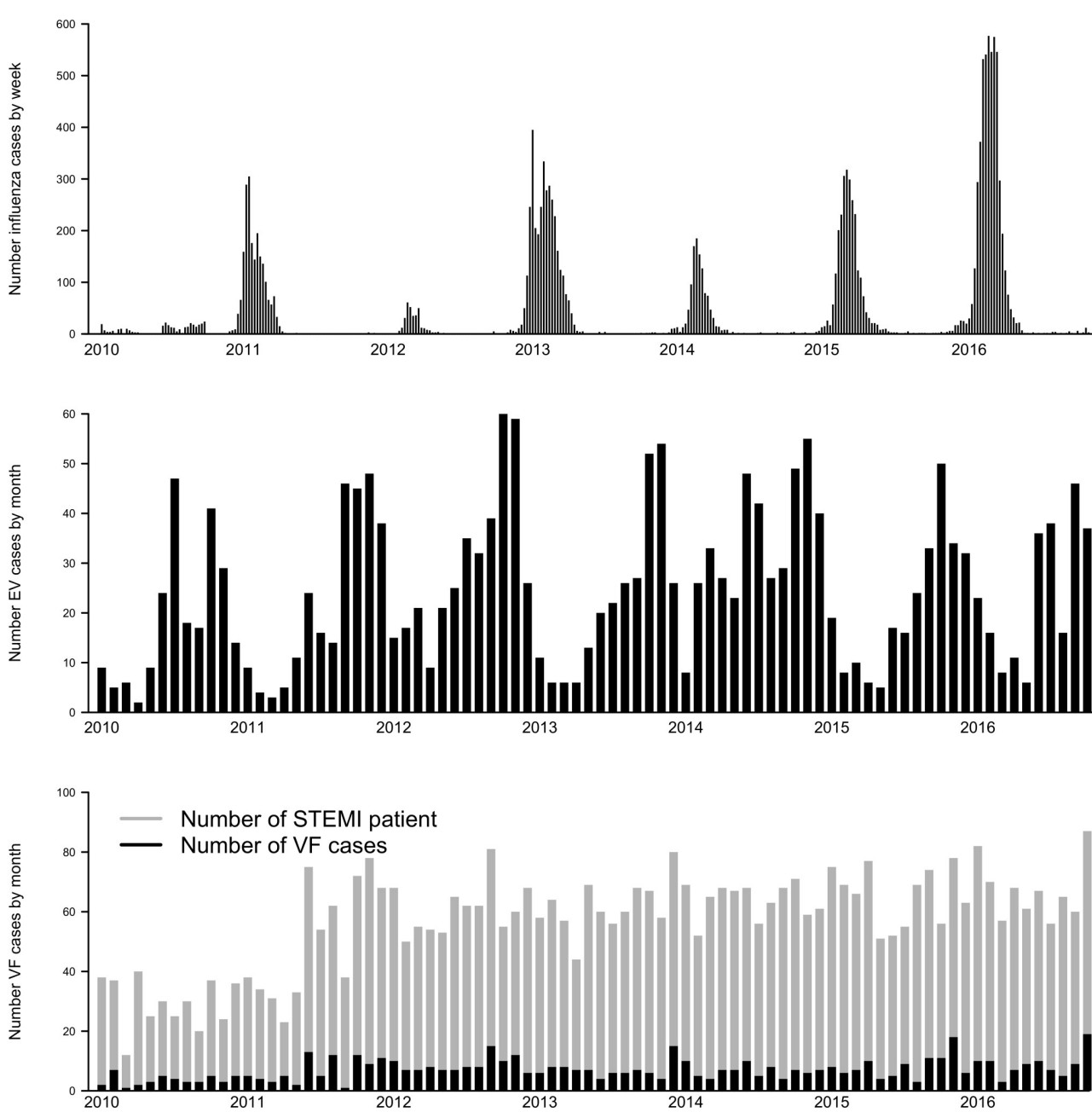

**Fig 2. Number of influenza, enterovirus (EV) and ventricular fibrillation (VF) cases per week and month by year.**

18–80 years), of whom 12% developed VF prior to PPCI. It was found that VF patients had experienced more generalized fatigue and flu-like symptoms within 7 days before STEMI compared with the patients without VF (OR 3.39, 95% CI 1.76–6.54). The expected winter peak and summer nadir of VF were not observed. After correlating Danish viral exposure surveillance data with individual data on VF risk among first STEMI, no such relationship was found between EV or influenza exposure and occurrence of VF. Therefore, there is no support for an association between higher levels of EV or influenza and high VF prevalence.

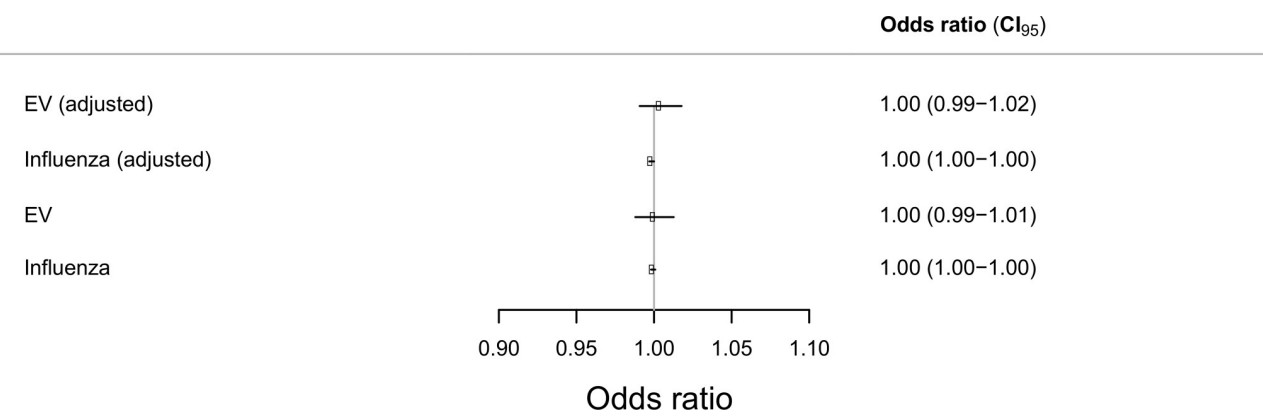

**Fig 3. Odds ratio for ventricular fibrillation during first ST-elevation myocardial infarction.**

## Seasonality of VF and association with viral infections

The seasonality of MI, ventricular arrhythmias, and SCA/SCD has for many decades been the focus of epidemiology studies, as seasonality may suggest etiological factors.[6–10] Because all patients in our study had a first STEMI, we were able to examine the seasonality and relationship of viral exposure with VF independently of its association with STEMI. Most,[8,32] but not all,[33] studies showed a seasonal variation of SCD, with a peak in winter (December and January) and a reduction during the summer months (from June through September). A link between viral infection of the heart (myocarditis) and sudden cardiac death has long been recognized.[34] At least 20 different viruses, including EV, have been implicated in myocarditis. [35] Although most patients infected with EV remain asymptomatic or only experience flu-like symptoms, the role of EV as one of the most common causes of viral myocarditis is well-established.[36] Evidence for a role of EV infection in the modulation of cardiac electrical function and pro-arrhythmia is the fact that myocarditis is one of the most common findings at autopsy in young adults who died of SCD,[37] and patients with serious ventricular arrhythmias and survivors of SCD had prevalent detection of EV RNA in their endomyocardial biopsies, suggesting a role of EV in ventricular arrhythmias and SCD[38]. Although we did not find an association between influenza and VF risk during STEMI, substantial evidence has been gathered to support the effect of influenza among MI patients.[13,39] These findings are consistent across studies using various measures for viral exposure, such as influenza activity at the population level through surveillance data,[40,41] as well as at the individual level with laboratory-confirmed infection[13]. The latter is preferable, and Kwong et al. found in a self-controlled case-series study that laboratory-confirmed influenza in the week before MI was associated with increased risk of MI.[13] Incidence ratio for acute MI during the 7-day risk interval compared with the control interval was 6.05 (95% CI 3.86–9.50). These data suggest that influenza-like illness can cause MI.[13]

In support of a role for viral infection in the modulation of VF susceptibility, a common variant genome-wide association study in the Dutch AGNES population, which compared individuals with and without VF in the setting of a first acute STEMI,[16] identified a VF susceptibility locus at chromosome 21q21 (rs2824292) close to the *CXADR* gene. While this observation is intriguing, the association was not replicated in two similar MI populations. [42,43] The lack of replication might have been due to the necessity of exposure to EV that may vary by location and time of the year. However, CAR protein has a long-recognized role as viral receptor in the pathogenesis of viral myocarditis,[44] a physiologic role for the receptor

in localization of connexin 45 at the intercalated disks of the cardiomyocytes in the atrioventricular node, and a role in conduction of the cardiac impulse [17,45,46].

Although there is substantial evidence supporting the role of influenza in CVD mortality and SCA/SCD,[15,47] we did not find an association between influenza and VF risk during STEMI. There are several possible reasons for this discrepancy in associations. First, many studies were designed to investigate overall influenza mortality or CVD mortality rather than SCA/SCD[47]. Second, most studies included different cardiac causes of SCA and SCD with consequent etiological heterogeneity, and no study has specifically investigated VF in the setting of an acute first MI. The role of influenza infection may be different with different underlying cardiac causes of SCA/SCD. Lastly, some studies used SCA/SCD as a surrogate of VF, and thus some cases of sudden death due to bradyarrhythmic or noncardiac causes may have been misclassified.

## Strength and limitations

There are several strengths of the current study. Our data collection process was comprehensive, and we were able to adjust the VF risk for multiple significant covariates. Despite these strengths, there were also some limitations that should be noted. First, we did not have individual viral exposure data. Second, it employed an ecological study design. As such, it was not possible to examine the exact exposure history of each individual. In addition, the community burden of viral infections is substantially underestimated in surveillance data, as most patients with EV or influenza do not seek medical care. However, because the underreporting is unlikely to vary over time, there is no evidence that this would substantially bias our results. Furthermore, the EV incidence was on a monthly basis, and an option could have been to interpolate the EV data as weekly incidences. However, the data points for EV were too scarce to obtain reliable incidence estimates. Third, there is a concern for recall bias regarding the symptoms prior to event. Patients were questioned about their symptoms during the admission to hospital. However, we have previously found that cases experience less angina within the last 12 months compared with controls.[30] Fourth, even though data were prospectively collected, we performed retrospective analyses. Fifth, patients who died outside of hospitals were not included, and the results may not be generalizable to patients who do not survive to reach the hospital. Lastly, although the strength of this study is the completeness of data via link between clinical data (the Eastern Danish Heart Registry) and Danish nationwide registries the validity of our data depends largely on the accuracy of coding. It has been documented that the coding of risk factors has high validity and specificity in the Danish National Patient Registry.[26]

In this study, we focused on seasonal viral infections, but several other CVD risk factors display seasonal variation. Obesity, increased fat intake, reduced physical activity, and higher blood pressure and serum cholesterol levels are more prevalent in winter. However, traditional risk factors including hypertension, hypercholesterolemia, and smoking were not independently associated with VF risk during STEMI in our cohort.[24,30] Future studies should base infection status on the individual STEMI patient to detect whether active infection might increase risk for VF during acute STEMI.

## Clinical implications

Due to low survival after VF and the risk of neurological deficits in patients who survive cardiac arrest, prevention is key in addressing this important public health problem. In this study, 14% of the VF cases experienced flu-like symptoms during the week before their sudden

cardiac arrest, and these symptoms may provide an opportunity for medical intervention to prevent some cases of VF during acute ischemia, especially in high-risk patients.

## Conclusion

There was no clear seasonality of VF during STEMI; after correlating Danish viral exposure surveillance data with individual data on VF risk among first STEMI, no relationship was found between EV or influenza exposure and occurrence of VF. Even though we found that VF patients experienced more generalized fatigue and flu-like symptoms within 7 days before STEMI compared with the patients without VF, our data do not support the hypothesis that higher levels of EV and influenza are associated with high VF prevalence.

## Author Contributions

**Conceptualization:** Charlotte Glinge, Thomas Engstrøm, Michael W. T. Tanck, Connie R. Bezzina, Jacob Tfelt-Hansen, Reza Jabbari.

**Data curation:** Charlotte Glinge, Mia Ravn Jacobsen, Niels Kjær Stampe, Reza Jabbari.

**Formal analysis:** Charlotte Glinge, Jeppe Ekstrand Halkjær Madsen, Thomas Gerds.

**Methodology:** Charlotte Glinge, Sofie E. Midgley, Michael W. T. Tanck, Mia Ravn Jacobsen, Niels Kjær Stampe, Ramona Trebbien, Thomas Gerds, Reza Jabbari.

**Project administration:** Charlotte Glinge, Reza Jabbari.

**Resources:** Thomas Engstrøm, Frants Pedersen, Lars Køber, Christian Torp-Pedersen, Thea Kølsen Fischer, Jacob Tfelt-Hansen, Reza Jabbari.

**Supervision:** Thomas Engstrøm, Michael W. T. Tanck, Connie R. Bezzina, Jacob Tfelt-Hansen, Reza Jabbari.

**Visualization:** Charlotte Glinge, Jeppe Ekstrand Halkjær Madsen, Niels Kjær Stampe.

**Writing – original draft:** Charlotte Glinge.

**Writing – review & editing:** Thomas Engstrøm, Sofie E. Midgley, Michael W. T. Tanck, Jeppe Ekstrand Halkjær Madsen, Frants Pedersen, Mia Ravn Jacobsen, Elisabeth M. Lodder, Nour R. Al-Hussainy, Niels Kjær Stampe, Ramona Trebbien, Lars Køber, Thomas Gerds, Christian Torp-Pedersen, Thea Kølsen Fischer, Connie R. Bezzina, Jacob Tfelt-Hansen, Reza Jabbari.

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
