## [Decision Letter · Decision Letter 0]

11 Dec 2019

Seasonality of Ventricular Fibrillation at First Myocardial Infarction and Association with Viral Exposure

PONE-D-19-29659

Dear Dr. Glinge,

We are pleased to inform you that your manuscript has been judged scientifically suitable for publication and will be formally accepted for publication once it complies with all outstanding technical requirements.

With kind regards,

Corstiaan den Uil

Academic Editor

PLOS ONE

Additional Editor Comments (optional):

Reviewers' comments:

Reviewer's Responses to Questions

**Comments to the Author**

1. Is the manuscript technically sound, and do the data support the conclusions?

Reviewer #1: Yes

2. Has the statistical analysis been performed appropriately and rigorously? 

Reviewer #1: Yes

3. Have the authors made all data underlying the findings in their manuscript fully available?

Reviewer #1: Yes

4. Is the manuscript presented in an intelligible fashion and written in standard English?

Reviewer #1: Yes

5. Review Comments to the Author

Reviewer #1: The authors should be congratulated for this excellent work. Even if the results are negative, the tested hypothesis is of major importance and those findings will be helpful for appropriately guide future research in the field. In the study, the authors examine the influence of seasonality and association of increased enterovirus and influenza activity in the community with the risk of ventricular fibrillation (VF) during first ST-elevation myocardial infarction. Based on a large STEMI cohort and on individual-level linkage of data from Danish nationwide registries, they identified all consecutive STEMI patients admitted to acute angiography and Primary PCI (PPCI) at Rigs Hospitalet Copenhagen between January 2010 and October 2016. The cohort where linked to information on monthly/weekly surveillance data on entro - and influenza virus from the Danish National Microbiology Database. Of 4659 consecutively enrolled STEMI patients, 581 (12%) had VF before PPCI. The VF patients experienced more fatigue and flu-like symptoms 7 days prior to the STEMI event than the non-VF patients from the cohort, but no significant association between enterovirus, week-number and influenza virus exposure

was found. Based on the results the authors conclude that there was no seasonality during STEMI, and that there were no relationship between enterovirus and influenza virus exposure and occurrence of VF before PPCI in STEMI patients. The topic is of great interest since SCA still accounts for more than half of cardiovascular mortality, and because acute coronary artery disease represents the majority of underlying causes. Methodology is robust, paper very well written. Although findings are negative they add substantially to the literature and undoubtedly will help others to go forward in this field.

6. PLOS authors have the option to publish the peer review history of their article (what does this mean?). If published, this will include your full peer review and any attached files.

Reviewer #1: No

---

## [Editor Report · Acceptance letter]

27 Dec 2019

PONE-D-19-29659 

Seasonality of Ventricular Fibrillation at First Myocardial Infarction and Association with Viral Exposure 

Dear Dr. Glinge:

I am pleased to inform you that your manuscript has been deemed suitable for publication in PLOS ONE. Congratulations! Your manuscript is now with our production department. 

With kind regards,

on behalf of

Dr. Corstiaan den Uil 

Academic Editor

PLOS ONE